# An SMA Transducer for Sensing Tactile Sensation Focusing on Stroking Motion

**DOI:** 10.3390/ma16031016

**Published:** 2023-01-22

**Authors:** Ryusei Oya, Hideyuki Sawada

**Affiliations:** 1Graduate School of Advanced Science and Engineering, Waseda University, Tokyo 169-8555, Japan; 2Faculty of Science and Engineering, Waseda University, Tokyo 169-8555, Japan

**Keywords:** shape-memory alloy wire, force sensing, superelasticity, tactile sensor, human tactile perception, texture sensing

## Abstract

The authors have developed a micro-vibration actuator using filiform SMA wire electrically driven by periodic electric current. While applying the SMA actuators to tactile displays, we discovered a phenomenon that the deformation caused by a given stress to an SMA wire generated a change in the electrical resistance. With this characteristic, the SMA wire works as a micro-force sensor with high sensitivity, while generating micro-vibration. In this paper, the micro-force sensing ability of an SMA transducer is described and discussed. Experiments are conducted by sliding the SMA sensor on the surface of different objects with different speeds, and the sensing ability is evaluated to be related with human tactile sensation.

## 1. Introduction

Virtual reality (VR) and augmented reality (AR) technologies have developed rapidly, and are expected to be widely employed as the ‘second world’ in a variety of scenarios such as the metaverse and new forms of entertainment and business communication. These technologies provide realistic experiences mainly by controlling visual and auditory information to be perceived by users’ various senses. The presentation of tactile sensation, which is recognized by our skin, is an important technology to enhance perceptions of reality in VR and AR worlds; however, such technologies are still in the research stage, and no commercially-available devices and systems have been introduced so far. If tactile senses could be controlled in a virtual world, users would be able to further enjoy highly immersive games in entertainment fields and to extend the means of flexible communication in various business opportunities. Tactile communication will also be able to open new medical care in tele-medicine and tele-surgery. In order to create a system that controls tactile senses, it will be necessary to quantitatively measure the tactile information. For dealing with visual and auditory information, commonly-used devices such as cameras, microphones and speakers are commercially available. However, no standard equipment to record and display tactile information has been made available thus far.

Tactile sensing is expected to be utilized in various industries, and has been actively studied so far. Typical sensors applicable in tactile sensing include capacitive sensors, piezoresistive sensors, optical methods, and piezoelectric devices [1,2,3]. Capacitive tactile sensors are designed to settle capacitors between two electrodes, so that the electrostatic capacity changes when force is applied and the distance between the electrodes changes [4,5]. The capacitive sensors have the advantages of high frequency response, precise load measurement, and wide dynamic range, but they tend to be susceptible to external noise. Piezoresistive sensors, on the other hand, are robust against electric noise, since they utilize the piezoresistive effect [6,7]. However, they require a high-voltage power source for driving, and the frequency response is comparatively low. Optical tactile sensors are operated by converting a load into light intensity or refractive index [8,9]. They have the advantages of high frequency response, precise load measurement, and wide dynamic range. However, they are constructed with larger peripheral systems with optical sensors. Piezoelectric tactile sensors are operated by converting a load into an electrical signal via a piezoelectric element [10,11]. The sensors have the advantage of high frequency response, but their spatial resolution is comparatively low. Other tactile sensing techniques using ultrasonic [12] and acceleration [13] elements have also been proposed recently, but technical difficulties such as high energy consumption and complex system structures have been reported when considering practical applications.

A shape-memory alloy (SMA) is well-known as an alloy that remembers its original shape. An SMA is able to memorize its shape as an original one in austenite phase at higher temperature. The SMA changes the state from austenite to martensite upon cooling. It is deformable by the application of external force in the martensite phase at lower temperature, and returns to its preliminary-remembered shape when heated to transit into austenite phase [14,15]. Shape-memory alloys are widely applied as thermo-mechanical actuators in devices such as shape-memory springs, thermo-responsive valves and catheters, and highly efficient energy converters in the biomedical [16,17], aerospace [18,19], automotive [20,21], robotics [22,23] and construction fields [24,25], as well as human-machine interfaces [26,27]. Owing to the characteristics of their light weight, compact size and generatable force, the SMA actuators are expected to be alternatives to conventional electronic actuators, electric motors, and pneumatic and hydraulic actuators. Shape-memory alloys can also be utilized as sensors and have been applied to temperature sensors [28,29], magnetic sensors [30,31], and strain sensors [32,33]. Their transformation speed is, however, comparatively slow, since the phase transformation between martensite phase and austenite phase is led by the temperature, which is conducted by giving heat to the material body or radiating heat to the surrounding environment.

A filiformed SMA wire with a diameter of 50–100 μm presents unique characteristics, swiftly responding to temperature changes related to the martensite and austenite phases. By applying weak current to an SMA wire, heat is generated by the internal electrical resistance, and the wire instantly shrinks up to 5% lengthwise. When the current is stopped and the temperature drops, it returns to its original length by heat radiating to the air. The SMA wire is thin and flexible enough to be cooled down right after the current stops, and it returns to the initial length as the temperature shifts from the austenite to the martensite phase. This means that the contraction and return of the SMA wire can be precisely controlled by a properly-prepared pulse current, and the phenomena is physically recognized as micro-vibration, having different frequencies and amplitudes.

The authors have developed a micro-vibration actuator electrically driven by periodic current, generated by a current control circuit for tactile displays. A vibration actuator is composed of a 5 mm-long SMA wire with a diameter of 50 μm, and micro-vibration with an amplitude of 1–2 μm is generated, which is perceived by the human body as various tactile sensations in accordance with the different frequencies. We employ pulse-width modulated (PWM) current to control the vibration mode of the SMA wire using a specially-designed current controller. The pulse has an amplitude of H volts, a width of W msec and a period of L msec. The duty ratio W/L determines the heating and cooling time of the SMA. The value H × W, which is equivalent to the calories exchanged, determines the amplitude of a vibration, and the vibration frequency is completely controlled by regulating L. The deformation of an SMA wire in motion was recorded by using a high-speed camera, and we verified that the SMA wire perfectly synchronized with the ON/OFF pulse current.

The authors have also discovered that the deformation caused by stress applied to an SMA wire generates a change in the electrical resistance. With this characteristic, the SMA wire works as a force sensor with high sensitivity, while generating micro-vibration. We have developed a sensor structure to effectively conduct a given stress to an SMA wire. In this paper, the micro-force-sensing ability of the SMA transducer is studied and discussed. Experiments are conducted by sliding the SMA transducer on the surface of different objects with different speeds, and the sensing ability will be evaluated so as to be related to human tactile sensations. The new transducer using SMA wires will be applied to tactile displays and sensors to measure and record various tactile sensations, and also to be effectively employed as physical feedback in VR and AR environments.

## 2. SMA Sensor and Tactile Sensing System

### 2.1. Physical Properties of SMAs

SMA has two typical physical properties related to body temperature and force: shape memory effect and superelasticity [34], respectively, as shown in Figure 1.

The shape memory effect presented by the background colored in blue is observed by exchanging heat in the body. An SMA in martensite is deformable by the application of load, and the shape returns to its original form by receiving heat to transit to austenite. The transition between martensite and austenite is reversible by the heat application to the body or the heat radiation from the body.

The superelasticity, on the other hand, is the phenomenon observed in the austenite phase as shown in the orange color. A loaded SMA in austenite deforms to transit to martensite, which is called the stress-induced martensite phase, and the strain is released by removing the load. This transformation exhibits the change of electrical resistance, and especially the transformation between austenite and R-phase shows the quick response in time.

### 2.2. Filiform SMAs for Micro-Vibration Actuators

The authors have developed a micro-vibration actuator using a filiformed SMA wire electrically driven by pulsed current. Figure 2 shows a vibration actuator composed of a 5 mm-long SMA wire with a diameter of 0.05 mm. By applying weak current to the alloy, the temperature rises to T_2_ due to the generated heat inside the wire body, and the alloy shrinks up to 5% lengthwise relative to the original length. When the current stops and the temperature drops to T_1_ due to the heat radiation, the alloy returns to its original length. Figure 3 shows the temperature characteristics of the SMA wire employed in this study having the specific temperatures T_1_ = 68 and T_2_ = 72 degrees Celsius [35].

The SMA wire is so thin that it rapidly cools after the current stops, and returns to its original length when the temperature shifts from T_2_ to T_1_. This means that the shrinkage and the return to initial length of the SMA wire can be controlled by the pulse current. By driving the SMA wire with properly-prepared pulse current, micro-vibration with an amplitude of several micrometers is generated, which is perceived by the human body as tactile sensation, although the vibration is invisible. We employ pulse-width modulated (PWM) current having different frequencies and duty ratios generated from a specially-designed current controller to control the vibration mode of the SMA wire. Through our studies so far, we discovered that the vibration frequencies were controlled up to 1 kHz under duty ratios of approximately 2–20%.

### 2.3. SMA Wires for Force Sensing

Owing to the superelasticity, the deformation caused by a given tensile stress to an SMA wire generates the change of the electrical resistance. By measuring the resistance change, the amount of stress applied to a wire can be estimated.

Figure 4 illustrates the sensor structure using an SMA wire, which consists of a 75 µm (diameter) × 3 mm (length) SMA wire and a 1.5 mm (diameter) × 5 mm (length) round-head pin made of stainless steel. The SMA wire used for the tactile sensor is BMF75 (Toki Corporation), and its physical properties are shown in Table 1. BMF75 is a one-way shape-memory alloy, and is composed of Ti-Ni-Cu, which is characterized by stable self-expanding properties. The tip of the pin is soldered at the middle of the SMA wire, so that force applied to the other end of the pin is efficiently conducted to the SMA wire, as shown in Figure 5. Two lead wires are connected to both ends of the SMA wire to measure the electrical resistance by employing a specially-designed control circuit. The circuit provides weak electrical current to the SMA wire to keep the SMA in the austenite phase. When force is applied to the metal pin, tensile force is conducted to the wire to induce the stress-induced martensite phase, which is measured as the change in the electrical resistance.

We also tested the fatigue of the SMA wire against tensile force, which was anticipated due to the sensor structure. A 10 cm wire was prepared, and one end was fixed to a metal frame. At the other end, weights were hung until the wire was cut off. We confirmed the SMA wire was cut off at the weight of 1.3 kg, and the wire also properly worked after applying a tensile force of 1 kg.

### 2.4. Structure of SMA Tactile Sensing System

We have constructed a tactile sensing system, as shown in Figure 6. The system consists of an SMA transducer mounted on a liner plotter (AxiDraw SE/A3), a transducer control circuit, and a deep learning-based classifier. In this experiment, we consider a situation where a human feels a tactile sensation of an object by sliding their fingertip on the object surface. When we try to feel an object by hand, we may place our fingers on the surface and stroke our hand with different pressures and speeds to properly understand the textures. Different objects have different textures with various softness, roughness and elasticity. To suitably perceive the textures, the touching pressure and the sliding speed may be suitably adjusted to understand the different physical structures on the surface.

To realize the human-like recognition of tactile sensation, the SMA sensor is mounted on a linear plotter to control the sliding speed, as shown in Figure 7. A weight of 100 g is placed on the actuator so that the pressure can be applied while sliding the sensor on an object surface. Since the SMA wire keeps its performance with the strain up to 1 kg, the sensor works properly with a 100 g weight.

The resistance value of the SMA wire *R_sma_* is measured by the following calculation: (1)$$V_{out}=3.0\cdot \frac{10}{10+R_{sma}}$$
using the circuit shown in Figure 8. Time-series data obtained by the sensing system is given to a classifier using the Transformer [36,37].

## 3. Preliminary Experiment

To find the sensing characteristics of the tactile transducer system, we firstly conducted a preliminary experiment. Five materials having different textures, namely a rubber mat, a plastic mesh, cotton fabric, a Velcro tape and a plastic sheet, were selected to test the sensing ability of the sensor. The relationship of the texture features is shown in Figure 9. For example, a rubber mat has a less bumpy surface with a periodic pattern, while by contrast, a Velcro tape is made up of random and rough surfaces. All the materials were prepared in a size of 30 cm × 40 cm, and were spread on a table. The SMA sensor unit mounted on the linear plotter stroked each material from one end to the other at eight different speeds, which were set at 1, 3, 5, 7.5, 10, 12.5, 15 and 20 cm/s. A weight of 100 g was placed on the sensor for applying initial stress to the SMA wire.

Sampling rate was set at 5000 Hz, and 50 stroking trials were conducted at each speed for each material. Figure 10 shows the examples of sensing data in one second obtained by changing the stroking speed. At each speed, different surfaces present different signal features, having particular periods and amplitudes. By changing the sliding speed, frequencies found in sensing data increase in accordance with the texture patterns and sliding speed. A plastic sheet, for example, has a smooth surface with low friction, and the obtained data have small amplitude without particular patterns. A plastic mesh, on the other hand, has a clear periodic pattern, and in the obtained data, clear repeating patterns are observed for all the different speeds.

## 4. Tactile Classification Using SMA Sensing System

In the preliminary experiment, we discovered that the SMA sensor was able to output reasonable signals by the stroking actions on different material surfaces. Here, we conducted tactile classification by using ten materials with different textures, including the five materials used in the preliminary experiment. Figure 11 shows the pictures of the material surfaces, which were a rubber mat, a plastic mesh, cotton fabric, a Velcro tape, a plastic sheet, a sheet made of bamboo, a plastic-made tile, a plastic mat with small patterns, a plastic mat with large patterns, and a leather sheet with studs.

For each material, the sliding action was conducted for 2 s, which was repeated 50 times. Sensing data were obtained with a sampling rate of 5 kHz, and the data were then down sampled into 100 Hz after the low-pass filtering. Data with a stroking-time duration of 500 ms were fed to the classifier. A deep learning-based classifier using Transformer was employed for the recognition of the materials. Eighty percent of the samples obtained from each material at each speed were used for the network training, and the other 20% of the samples were used for the validation.

Classification results at the sliding speed of 10 cm/s are summarized as a confusion matrix in Figure 12. The horizontal axis of the confusion matrix is the predicted result for each material obtained by the discriminator, and the vertical axis represents the correct label for each material. The average recognition rate for the ten materials was 86.2%. The materials with distinctive textures, such as the Velcro tape, bamboo sheet and plastic tile, presented high accuracy of 100%. On the other hand, the materials having similar smoothness, such as the rubber mat and the two plastic mats, were confused with one another. Since the diameter of the tactile pin is 1.5 mm, the size of the pin tip is smaller than the texture patterns of the mats, as shown in Figure 13. When the pin tip contacted the edge of the texture patterns, similar signals were output for those three materials, which might have given greater effect to the classification.

To avoid such misrecognition, we will consider the shape of the pin tip and the pin material to improve the classification ability. The softness of human skin, together with the structure of fingerprints, are considered to be important in how the perception of tactile sensation is processed. By referring to the skin structure, our tactile sensor will be improved in terms of how it conducts the physical stimuli to the SMA wire.

## 5. Conclusions

In this paper a tactile transducer using an SMA wire was introduced. We discovered that the resistance change against the force application to an SMA wire was caused by the superelasticity, and was given as the physical property to the tactile sensing. By introducing the special structure employing a tactile pin, the physical stimuli obtained by the sliding motion on a material surface was efficiently conducted to the SMA wire to cause the resistance change. By examining the output signals, the characteristic patterns were obtained from different material surfaces, in accordance with the stroking speed. We also employed the deep learning-based classifier, and verified the classification performance against ten different textures. An average of 86.2% recognition rate was achieved. However, classification performance was slightly decreased among the materials that had similar textures. We will further improve the shape of the tactile pin, together with the compositing material of the pin.

## Figures and Tables

**Figure 1 materials-16-01016-f001:**
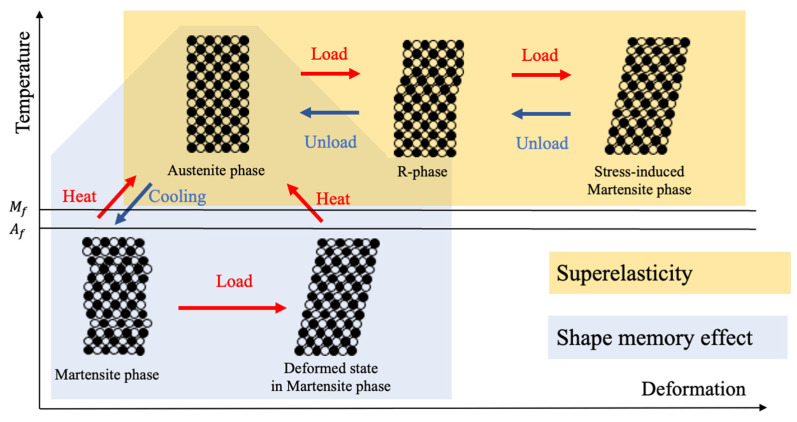
Physical properties of the deformation of an SMA.

**Figure 2 materials-16-01016-f002:**
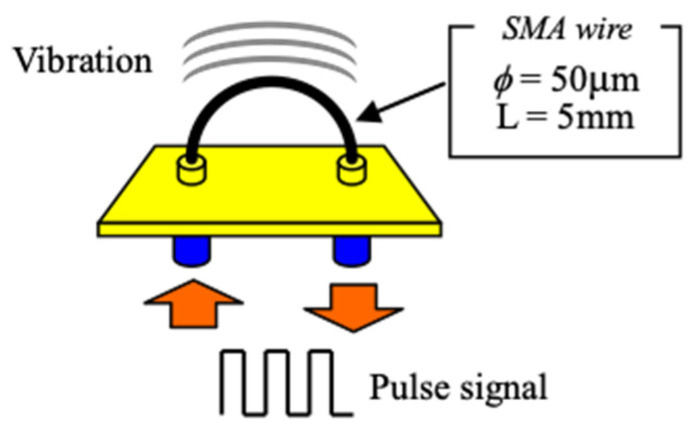
SMA vibration actuator.

**Figure 3 materials-16-01016-f003:**
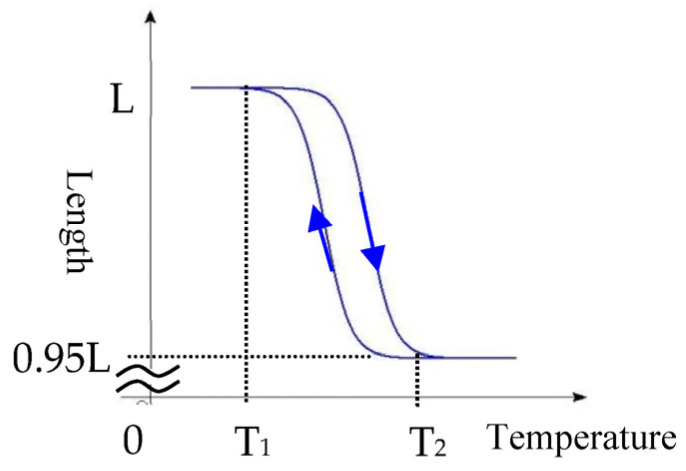
Temperature characteristics of SMA wire.

**Figure 4 materials-16-01016-f004:**
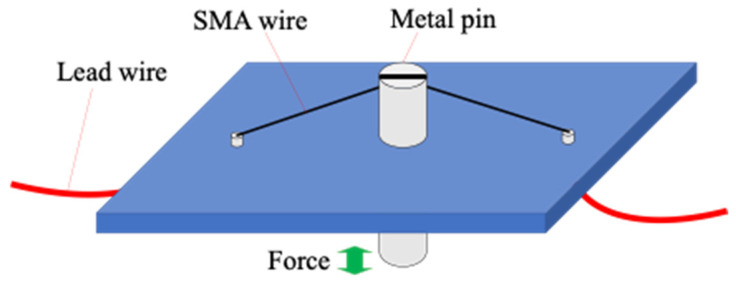
SMA sensor for sensing micro-force.

**Figure 5 materials-16-01016-f005:**
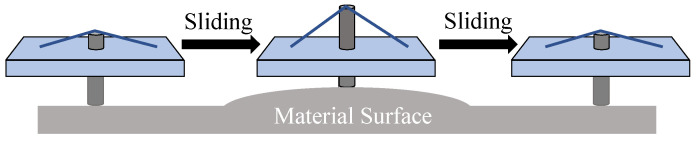
Sliding the SMA sensor on material surface.

**Figure 6 materials-16-01016-f006:**
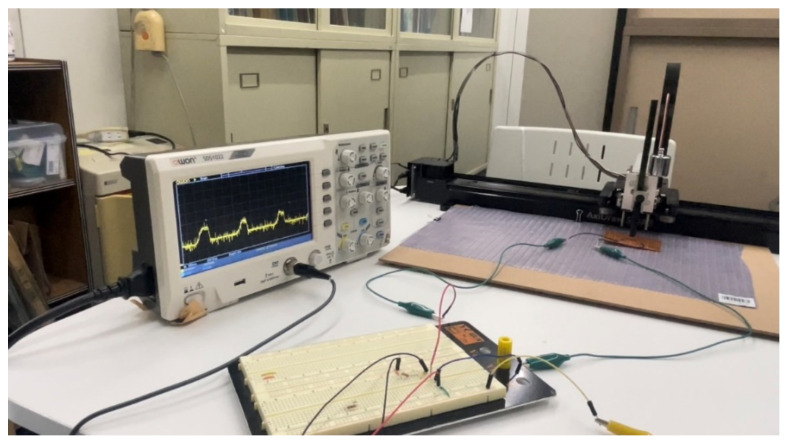
Tactile sensing system using a SMA sensor.

**Figure 7 materials-16-01016-f007:**
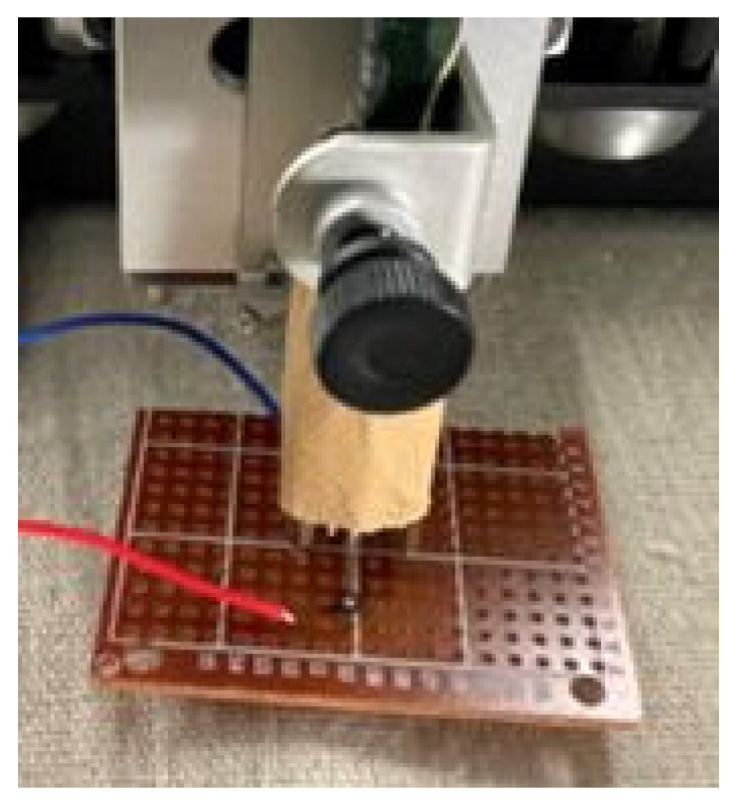
SMA sensor mounted on a linear plotter.

**Figure 8 materials-16-01016-f008:**
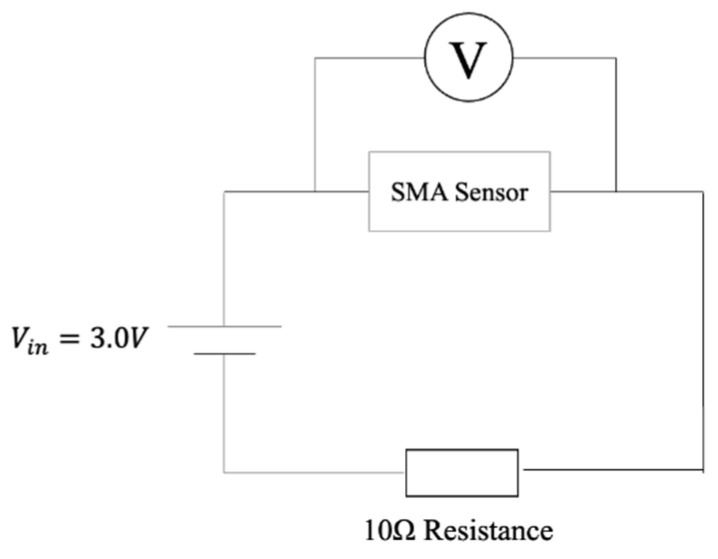
SMA driving circuit.

**Figure 9 materials-16-01016-f009:**
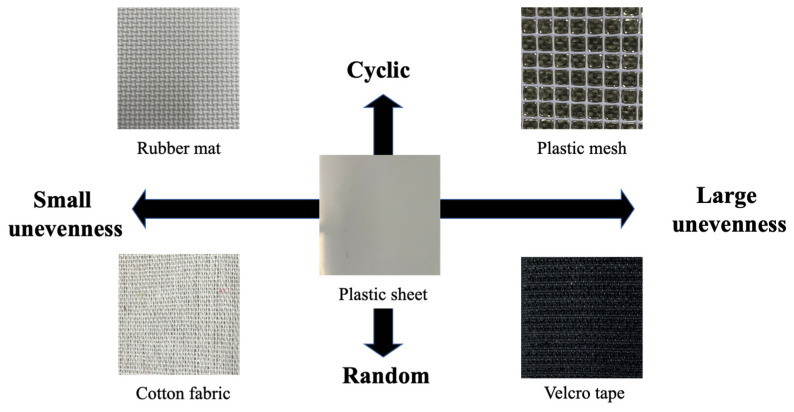
Selected five materials with different textures.

**Figure 10 materials-16-01016-f010:**
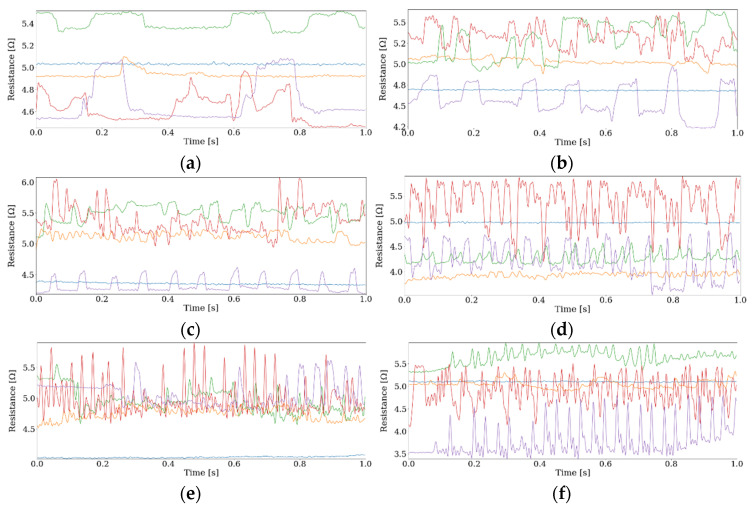
Examples of sensing data with different stroking speeds: (**a**) sliding speed 1 cm/s, (**b**) sliding speed 3 cm/s, (**c**) sliding speed 5 cm/s, (**d**) sliding speed 7.5 cm/s, (**e**) sliding speed 10 cm/s, (**f**) sliding speed 12.5 cm/s, (**g**) sliding speed 15 cm/s, (**h**) sliding speed 20 cm/s.

**Figure 11 materials-16-01016-f011:**
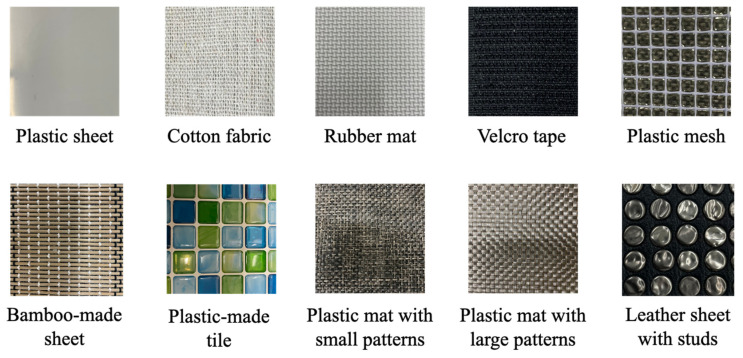
Ten materials for tactile classification experiment.

**Figure 12 materials-16-01016-f012:**
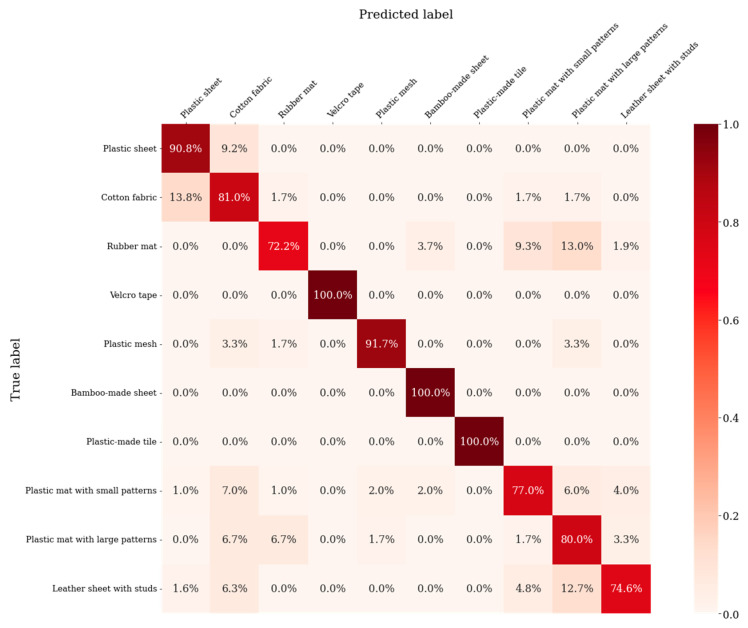
Confusion matrix of classification for ten different materials at sliding speed 10 cm/s.

**Figure 13 materials-16-01016-f013:**
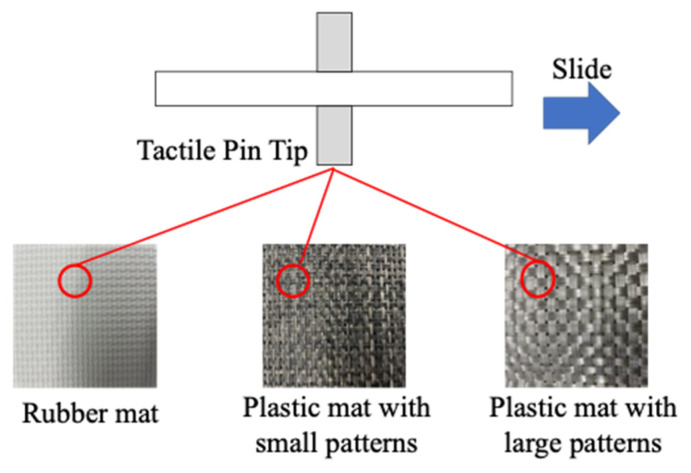
Contact situation between a tactile pin tip and material surface.

**Table 1 materials-16-01016-t001:** Physical properties of BMF75.

Physical Property	Value
Standard diameter (µm)	75
Practical force produced (load) (gf)	35
Practical kinetic strain (%)	4.0
Standard drive current (mA)	140
Standard drive voltage (V/m)	35.4
Standard power (W/m)	4.63
Standard resistance (Ω/m)	236
Tensile strength (Kgf)	0.45
Weight (mg/m)	28

## Data Availability

Not applicable.

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
