# Peer review of "An SMA Transducer for Sensing Tactile Sensation Focusing on Stroking Motion"

_materials, 2023, doi:10.3390/ma16031016_

Round 1

Reviewer 1 Report

Review of the manuscript "An SMA Transducer for Sensing Tactile Sensation Focusing on Stroking Motion". A micro-vibration actuator using filiformed SMA wire electrically driven by periodic electric current has been proposed by the authors.

The manuscript is original, and the idea interesting. English should be revised by english native-speaker:

example:

Line 174: to keep NOT to keeps.

Line 196: linear NOT liner.

In my opinion it can be reconsidered for publication after the following major revisions:

1) References list is not in compliance with the journal requirements. Please follow strictly auhtor's instructions.

2) References are quite ancient. Please add more recent articles.

2) Results (Fig. 12) reported as confusion matrix are quite... confusing. At least it should be introduced the basic concept of such data. Furthermore a different presentation of the results is warmly recommended.

After these changes the manuscript can be reconsidered for publication.

Reviewer 2 Report

In manuscript titled “An SMA transducer for sensing tactile sensation focusing on stroking motion”, submitted to the journal `Materials’, the authors discuss about developing a mico-actuator where applying mechanical force on the SMA wire causes change in electrical resistance. The SMA wire is used as a micro force sensor, and its performance is tested on several materials. The manuscript is within the scope of the journal, it is well-written, and can be published after the authors clarify the following queries and incorporate the suggested changes:

1.     The first two lines in the abstract `A shape-memory alloy …. into austenite phase’ is unnecessary within the abstract. They should be removed.    

2.     Nowhere in the entire manuscript the exact SMA used for the sensor is not mentioned. The composition of the SMA along with the kind of phase transformations it undergoes should be mentioned. The transformation strains, which will decide the extension of the wire, also should be specified.

3.     A comment about the applicability of various other SMAs will be useful.

4.     While describing the microstructure evolution, a book on microstructure should be cited. For example, the one by Kaushik Bhattacharya.                                                     

Round 2

Reviewer 1 Report

The manuscript has been improved and it can be accepted in the present form

Reviewer 3 Report

accept